# An Improved Approach for Implementing Dynamic Mode Decomposition with Control

Gyurhan Nedzhibov 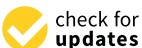

Faculty of Mathematics and Informatics, Shumen University, 9700 Shumen, Bulgaria; g.nedzhibov@shu.bg

**Abstract:** Dynamic Mode Decomposition with Control is a powerful technique for analyzing and modeling complex dynamical systems under the influence of external control inputs. In this paper, we propose a novel approach to implement this technique that offers computational advantages over the existing method. The proposed scheme uses singular value decomposition of a lower order matrix and requires fewer matrix multiplications when determining corresponding approximation matrices. Moreover, the matrix of dynamic modes also has a simpler structure than the corresponding matrix in the standard approach. To demonstrate the efficacy of the proposed implementation, we applied it to a diverse set of numerical examples. The algorithm's flexibility is demonstrated in tests: accurate modeling of ecological systems like Lotka-Volterra, successful control of chaotic behavior in the Lorenz system and efficient handling of large-scale stable linear systems. This showcased its versatility and efficacy across different dynamical systems.

**Keywords:** dynamic mode decomposition with control; DMDc method; DMD method; Koopman operator; equation-free

## 1. Introduction

Dynamic Mode Decomposition with Control (DMDc) is a powerful data analysis and modeling technique that has gained significant attention in various scientific and engineering disciplines. The method was introduced by Proctor et al. [1] and it provides valuable insights into the behavior of complex dynamical systems under the influence of external control inputs or actuation signals. The ability to understand and predict the behavior of such systems is of paramount importance in diverse applications, including biology [1], fluid dynamics and structural mechanics [2], aerodynamic forces [3], aerospace structures [4], aircraft parameter estimation [5] and wind farm flow control [6]. There are a number of connections between DMDc and other popular system identification methods such as the Eigensystem Realization Algorithm (ERA), the Observer Kalman Identification Method (OKID) and the Koopman operator with control, see [7–9]. For some more recent results and applications, we refer the reader to [10–14].

Many real-world systems are subject to external control forces or inputs. The mentioned publications address practical and relevant problems and the application of the DMDc method to solve them. Although they demonstrate the capabilities of the DMDc method for the study of controlled systems in these different areas, many open questions remain. Like traditional DMD, DMD with control relies heavily on high-quality, high-dimensional data. Gathering such data can be expensive and time-consuming and the method's effectiveness may be limited by the availability and quality of data. A number of questions are of great importance, including assumptions made during the modeling process, data collection methods, potential sources of error, or the impact of data quality on the model's predictions. The discussion of issues such as dealing with noise in measurements, choosing appropriate control inputs and addressing the issue of dimensionality when working with high-dimensional data is relatively rare. All these questions open new avenues for future analysis and research.

The DMDc method is an extension of the Dynamic Mode Decomposition (DMD method) [15], which has been established as a method for analyzing data from numerical simulations and laboratory experiments in the fluid dynamics field. The method constitutes a mathematical technique for identifying spatiotemporal coherent structures from high-dimensional data. Since its introduction, the method has been used in a variety of fields, including video processing [16], epidemiology [17], robotics [18], neuroscience [19], financial trading [20–22], cavity flows [23,24], and various jets [25,26]. For a review of the DMD literature, we refer the reader to [2,8,27,28]. The modeling of a high-dimensional, complex, dynamical system often requires external input or control. The standard DMD does not take this control into account and will not always predict the system correctly. On the other hand, DMDc utilizes both the measurements of the system and the applied external control to extract the underlying dynamics.

In this paper, we present a novel and efficient algorithm for implementing DMDc, addressing a key challenge in the field of computational analysis. Our approach aims to significantly improve the computational efficiency of DMDc without compromising its predictive accuracy, making it an attractive tool for real-time applications and large-scale systems. The contribution of our work lies in the development of a streamlined algorithm that seamlessly integrates control inputs into the dynamic mode decomposition process. By leveraging singular value decomposition of lower order matrices and a simpler structure of the approximation matrices, our approach significantly reduces the computational overhead associated with the traditional DMDc method. This advancement is crucial, as computational efficiency is often a critical concern in practical applications, where real-time analysis and control are required. This efficiency breakthrough opens up new possibilities for studying complex systems with control in real-time scenarios, providing valuable insights for researchers and engineers alike. In summary, our work contributes to the growing body of research on dynamic mode decomposition with control, offering a novel and efficient computational framework that advances the field's capabilities. The combination of computational efficiency and predictive accuracy demonstrated through numerical experiments makes our approach a promising tool for a wide range of scientific and engineering applications. We believe that our work will inspire further investigations and applications in the exciting and interdisciplinary realm of DMDc, pushing the boundaries of understanding and controlling complex dynamical systems.

The outline of the paper is as follows: in Section 2, we briefly describe the DMDc algorithm; in Section 3, we propose and discuss the new approache for DMDc computation; and in Section 4, we present numerical results, while the conclusion is in Section 5.

## 2. Problem Statement

DMDc is a recent extension of the DMD method developed to analyze complex systems that also have inputs and disturbances [1]. The method uses both system measurements and applied external control inputs to extract the underlying dynamics and the impact of the inputs. We assume that there is no noise or uncertainties in the observations.

Consider a sequential set of data arranged in a $n \times m + 1$ matrix

$$Z = [\mathbf{x}_0, \ldots, \mathbf{x}_m] \tag{1}$$

where $n$ is the number of state variables and $m + 1$ is the number of observations (*snapshots*). The data $\mathbf{x}_i$ could be from measurements, experiments, or simulations collected at the time $t_i$ from a given dynamical system, assuming that the data are equispaced in time with a time step $\triangle t$. To proceed, we use an arrangement of the dataset into two large data matrices

$$X = [\mathbf{x}_0, \ldots, \mathbf{x}_{m-1}] \quad \text{and} \quad Y = [\mathbf{x}_1, \ldots, \mathbf{x}_m]. \tag{2}$$

The time-varying inputs to the system are collected to form another data matrix

$$\Gamma = [\mathbf{u}_0, \ldots, \mathbf{u}_{m-1}], \tag{3}$$

where $\mathbf{u}_i \in \mathbf{R}^q$ and $q$ denotes the number of input variables.

The ranks of matrices $X$, $Y$ and $\Gamma$ are subject to certain conditions in DMDc. It is typical for the rank of matrix $Y$ to be the same as the rank of matrix $X$ because both are constructed from the same snapshots of data. It depends on the dimensionality and complexity of the dynamic system being analyzed. The rank of the control input data matrix $\Gamma$ depends on the nature of the control inputs and the specific dataset. In practice, we can analyse the structure of $X$ and $\Gamma$ and determine their effective rank by using techniques such as SVD or other dimensionality reduction methods.

The goal of DMDc is to find a relationship between the future state $\mathbf{x}_{k+1}$, the current state $\mathbf{x}_k$ and the external control inputs $\mathbf{u}_k$, given by the following relation

$$\mathbf{x}_{k+1} = A\mathbf{x}_k + B\mathbf{u}_k, \tag{4}$$

where $A \in R^{n \times n}$ is called the DMD operator and $B \in R^{n \times q}$ is called the input matrix. The relation (4) can be rewritten in terms of the snapshot matrices

$$Y = AX + B\Gamma. \tag{5}$$

We assume that both operators $A$ and $B$ are unknown and that the goal is to obtain estimates of matrices $A$ and $B$.

## 3. Dynamic Mode Decomposition with Control (DMDc)

The system description (5) can therefore be rewritten in an augmented form as

$$Y = \begin{bmatrix} A & B \end{bmatrix} \begin{bmatrix} X \\ \Gamma \end{bmatrix} = G\Omega, \tag{6}$$

where

$$G = [A\,B] \in \mathbf{R}^{n \times (n+q)} \quad \text{and} \quad \Omega = \begin{bmatrix} X \\ \Gamma \end{bmatrix} \in \mathbf{R}^{(n+q) \times m}. \tag{7}$$

The Dynamic Mode Decomposition with control of the measurement trio $(Y, X, \Gamma)$ is the eigendecomposition of the operator $A$.

We seek a best-fit solution for the operator $G$. The matrix of known terms $\Omega$ can be approximated via the singular value decomposition (SVD): $\Omega = U\Sigma V^*$. This expression may be truncated to the form

$$\Omega \approx \tilde{U}\tilde{\Sigma}\tilde{V}^*, \tag{8}$$

where $\tilde{U} \in \mathbf{R}^{(n+q) \times p}$, $\tilde{\Sigma} \in \mathbf{R}^{p \times p}$ and $\tilde{V} \in \mathbf{R}^{m \times p}$, which gives the approximation

$$G \approx Y\tilde{V}\tilde{\Sigma}^{-1}\tilde{U}^*. \tag{9}$$

We can represent the matrix $\tilde{U}$ as follows

$$\tilde{U} = \begin{bmatrix} \tilde{U}_1 \\ \tilde{U}_2 \end{bmatrix}, \tag{10}$$

where $\tilde{U}_1 \in \mathbf{R}^{n \times p}$, $\tilde{U}_2 \in \mathbf{R}^{q \times p}$. Then, we can approximate the matrices $A$ and $B$ in the following manner

$$A \approx \bar{A} = Y\tilde{V}\tilde{\Sigma}^{-1}\tilde{U}_1^* \quad \text{and} \quad B \approx \bar{B} = Y\tilde{V}\tilde{\Sigma}^{-1}\tilde{U}_2^*. \tag{11}$$

In the next stage, we obtain reduced representations of the dynamics $A$ and the input matrix $B$. For this purpose, we project the state onto a subspace on which it evolves using a basis transformation. In DMDc, in contrast with DMD, the matrix of truncated left singular

vectors $\tilde{U}$ cannot be used to define this transformation. In order to find the appropriate linear transformation, we utilize a reduced-order SVD of $Y$

$$Y \approx \hat{U}\hat{\Sigma}\hat{V}^*, \tag{12}$$

where $\hat{U} \in \mathbf{R}^{n \times r}$, $\hat{\Sigma} \in \mathbf{R}^{r \times r}$ and $\hat{V} \in \mathbf{R}^{m \times r}$, and $r$ denotes the dimension of the subspace. In general, the dimension $p$ of the reduced SVD for $\Omega$ is greater than the dimension $r$. Using the transformation $\hat{U}$, a low-dimensional representation of the matrices $A$ and $B$ can be computed as follows:

$$\tilde{A} = \hat{U}^* \bar{A} \hat{U} = \hat{U}^* Y \tilde{V} \tilde{\Sigma}^{-1} \tilde{U}_1^* \hat{U},$$
$$\tilde{B} = \hat{U}^* \bar{B} = \hat{U}^* Y \tilde{V} \tilde{\Sigma}^{-1} \tilde{U}_2^*, \tag{13}$$

where $\tilde{A} \in \mathbf{R}^{r \times r}$ and $\tilde{B} \in \mathbf{R}^{r \times q}$.

We should note that in order to obtain the necessary basis transformation, we can equivalently use the singular value decomposition of matrix $X$. The corresponding linear transformation will be unique depending on the choice of $X$ or $Y$.

In a similar way to the DMD approach, we can obtain the dynamic modes of $A$ by the eigendecomposition

$$\tilde{A}W = W\Lambda. \tag{14}$$

The eigenvectors of $A$ are related to the eigenvectors of $\tilde{A}$ via the following transformation

$$\Phi = Y\tilde{V}\tilde{\Sigma}^{-1}\tilde{U}_1^* \hat{U}W, \tag{15}$$

where the columns of the matrix $\Phi$ are the DMD modes.

The Algorithm 1 detailing the application of the DMDc method is presented below.

---

**Algorithm 1:** DMDc algorithm [1]

1. Collect and construct the snapshot matrices: $X, Y$ and $\Gamma$

   as defined in (2) and (3). Construct the matrix $\Omega$ as in (7).

2. Compute the truncated SVD of $\Omega$

   $\Omega = \tilde{U}\tilde{\Sigma}\tilde{V}^*,$

   with truncation value $p$.

3. Compute the truncated SVD of $Y$

   $Y = \hat{U}\hat{\Sigma}\hat{V}^*,$

   with truncation value $r$.

4. Compute the reduced approximations of $A$ and $B$ :

   $\tilde{A} = \hat{U}^* \bar{A} \hat{U} = \hat{U}^* Y \tilde{V} \tilde{\Sigma}^{-1} \tilde{U}_1^* \hat{U},$

   $\tilde{B} = \hat{U}^* \bar{B} = \hat{U}^* Y \tilde{V} \tilde{\Sigma}^{-1} \tilde{U}_2^*,$

5. Perform the eigenvalue decomposition of $\tilde{A}$

   $\tilde{A}W = W\Lambda.$

6. Compute the DMD modes of $A$

   $\Phi = Y\tilde{V}\tilde{\Sigma}^{-1}\tilde{U}_1^* \hat{U}W.$

---

## 4. Alternative and Improved DMDc Algorithms

This subsection introduces a novel approach to the DMDc method, which enhances efficiency over the standard approach. It was partly introduced in our recent conference report [29]. For completeness of the exposition, we will describe this novel methodology below. It is an additional approach that may be of interest to readers seeking a broader understanding of the problem domain.

### 4.1. An Alternative to the DMDc Algorithm

As in (2) and (3), let us define the snapshot matrices $X, Y$ and $\Gamma$. We define the matrix $\Omega$ as in (7). The best-fit linear operator $G$, in (7), is approximated by

$$G = Y\Omega^\dagger, \tag{16}$$

where $\Omega^\dagger$ is the Moore–Penrose pseudoinverse of $\Omega$. We can use different approaches to calculate $\Omega^\dagger$. For example, in the case where $\Omega$ has linearly independent rows, $\Omega^\dagger$ can be computed as $\Omega^*(\Omega\Omega^*)^{-1}$. Let us represent the matrix $\Omega^\dagger$ as follows

$$\Omega^\dagger = \begin{bmatrix} \Omega_1 & \Omega_2 \end{bmatrix}, \tag{17}$$

where $\Omega_1 \in \mathbf{R}^{m \times n}$ and $\Omega_2 \in \mathbf{R}^{m \times q}$. Then, we can approximate the matrices $A$ and $B$ in the following manner

$$A \approx \check{A} = Y\Omega_1 \text{ and } B \approx \check{B} = Y\Omega_2. \tag{18}$$

If we use a truncated SVD of $\Omega$ as in (8), then matrices $\check{A}$ and $\check{B}$ coincide with $\bar{A}$ and $\bar{B}$ in (11), respectively. In order to obtain a reduced representation of the dynamics $A$ and the input matrix $B$, we utilize a reduced-order SVD of $\Omega_1$ instead of $Y$. Let us write

$$\Omega_1 \approx \check{U}\check{\Sigma}\check{V}^*, \tag{19}$$

where $\check{U} \in \mathbf{R}^{m \times r}$, $\check{\Sigma} \in \mathbf{R}^{r \times r}$ and $\check{V} \in \mathbf{R}^{n \times r}$ and $r$ denotes the dimension of the subspace. Using the transformation matrix $\check{V}$, a low-dimensional representation of the matrices $A$ and $B$ can be computed as follows:

$$\begin{aligned} \tilde{A} &= \check{V}^*\check{A}\check{V} = \check{V}^*Y\check{U}\check{\Sigma}, \\ \tilde{B} &= \check{V}^*\check{B} = \check{V}^*Y\Omega_2, \end{aligned} \tag{20}$$

where $\tilde{A} \in \mathbf{R}^{r \times r}$ and $\tilde{B} \in \mathbf{R}^{r \times q}$. Note that $\tilde{A}$ and $\tilde{B}$ in (20) differ from the corresponding matrices in (13). We can obtain the dynamic modes of $A$ by the eigendecomposition of $\tilde{A}$

$$\tilde{A}W = W\Lambda. \tag{21}$$

The eigenvectors of $A$ are related to the eigenvectors of $\tilde{A}$ via the transformation

$$\Phi = Y\check{U}\check{\Sigma}W, \tag{22}$$

where the columns of the matrix $\Phi$ are the DMD modes.

Next, we resume the results from above in the following Algorithm 2.

### 4.2. Improved DMDc Algorithms

In this subsection, we will introduce a new and improved alternative to the DMDc algorithm. We aim to minimize the computational overhead associated with Algorithm 2.

Let the snapshot matrices $X, Y$ and $\Gamma$ be defined as in (2) and (3), respectively, and let the augmented matrix $\Omega$ be defined as in (7).

Recall (8), the truncated singular value decomposition of $\Omega$

$$\Omega \approx \tilde{U}\tilde{\Sigma}\tilde{V}^*,$$

where $\tilde{U} \in \mathbf{R}^{(n+q) \times p}$, $\tilde{\Sigma} \in \mathbf{R}^{p \times p}$ and $\tilde{V} \in \mathbf{R}^{m \times p}$. Then, for the matrix representation $\Omega^\dagger$, from (17), we obtain

$$\Omega^\dagger = \begin{bmatrix} \Omega_1 & \Omega_2 \end{bmatrix} = \begin{bmatrix} \tilde{V}\tilde{\Sigma}^{-1}\tilde{U}_1^* & \tilde{V}\tilde{\Sigma}^{-1}\tilde{U}_2^* \end{bmatrix}, \tag{23}$$

where $\Omega_1 \in \mathbf{R}^{m \times n}$, $\Omega_2 \in \mathbf{R}^{m \times q}$ and matrices $\tilde{U}_1$, $\tilde{U}_2$ are defined by (10).

---

**Algorithm 2:** Alternative DMDc algorithm

---

1. Collect and construct the snapshot matrices: $X, Y$ and $\Gamma$
   as defined in (2) and (3). Construct the matrix: $\Omega$ as in (7).

2. Compute the pseudoinverse of $\Omega$
   $$\Omega^\dagger = \begin{bmatrix} \Omega_1 & \Omega_2 \end{bmatrix}.$$

3. Compute the truncated SVD of $\Omega_1$
   $$\Omega_1 = \check{U}\check{\Sigma}\check{V}^*,$$
   with truncation value $r$.

4. Compute the reduced approximations of $A$ and $B$:
   $$\tilde{A} = \check{V}^*\check{A}\check{V} = \check{V}^*Y\check{U}\check{\Sigma},$$
   $$\tilde{B} = \check{V}^*\check{B} = \check{V}^*Y\Omega_2,$$

5. Perform the eigenvalue decomposition of $\tilde{A}$
   $$\tilde{A}W = W\Lambda.$$

6. Compute the DMD modes of $A$
   $$\Phi = Y\check{U}\check{\Sigma}W.$$

---

For the sake of convenience, let us denote

$$\Omega_1 = \tilde{V}\tilde{H}, \tag{24}$$

where $\tilde{H} = \tilde{\Sigma}^{-1}\tilde{U}_1^{\,*}$ is the $p \times n$ matrix. We utilize the reduced-order SVD of $\tilde{H}$

$$\tilde{H} \approx \bar{U}\check{\Sigma}\check{V}^*, \tag{25}$$

where $\bar{U} \in \mathbf{R}^{p \times r}$, $\check{\Sigma} \in \mathbf{R}^{r \times r}$ and $\check{V} \in \mathbf{R}^{m \times r}$ and $r$ denotes the dimension of the subspace ($r \leq p$). Substituting (25) into (24), we obtain

$$\Omega_1 = \tilde{V}\bar{U}\check{\Sigma}\check{V}^* = \check{U}\check{\Sigma}\check{V}^*, \tag{26}$$

where $\check{U} = \tilde{V}\bar{U}$. It can easily be shown that

$$\check{U}^*\check{U} = I,$$

which implies that (26) is the truncated SVD of $\Omega_1$, matching (19).

To be thorough, we will prove that matrix

$$\Phi = Y\check{U}\check{\Sigma}W$$

corresponds to the exact DMD modes of $A$.

**Theorem 1.** *Let $(\lambda, \mathbf{w})$, with $\lambda \neq 0$, be an eigenpair of $\tilde{A}$ defined by (20). Then the corresponding eigenpair of $A$ is $(\lambda, \phi)$, where*

$$\phi = Y\check{U}\check{\Sigma}\mathbf{w}. \tag{27}$$

**Proof.** Let us express $A\phi$ by using (18), (19) and (27)

$$A\phi = Y\Omega_1\phi = Y\check{U}\check{\Sigma}\check{V}^*\phi = Y\check{U}\check{\Sigma}\check{V}^*Y\check{U}\check{\Sigma}\mathbf{w},$$

which implies, by using (20), that

$$A\phi = Y\check{U}\check{\Sigma}\tilde{A}\mathbf{w}.$$

Now, suppose that $\tilde{A}\mathbf{w} = \lambda\mathbf{w}$ for $\lambda \neq 0$. Then

$$A\phi = \lambda Y\check{U}\check{\Sigma}\mathbf{w} = \lambda\phi.$$

In addition, $\phi \neq 0$, since if $Y\check{U}\check{\Sigma}\mathbf{w} = 0$, then $\check{V}^*Y\check{U}\check{\Sigma}\mathbf{w} = \tilde{A}\mathbf{w} = 0$, which implies $\lambda = 0$. Hence, $\phi$ is an eigenvector of $A$ with eigenvalue $\lambda$.    $\square$

Using (24), (25) and (26), we can modify Algorithm 2 as follows (Algorithm 3):

---

**Algorithm 3:** Improved DMDc algorithm

---

1. Collect and construct the snapshot matrices: $X, Y$ and $\Gamma$
   as defined in (2) and (3). Construct the matrix: $\Omega$ as in (7).
2. Compute the pseudoinverse of $\Omega$ by truncated SVD $\Omega \approx \tilde{U}\tilde{\Sigma}\tilde{V}^*$
   with truncation value $p$ :
   $\Omega^\dagger = \begin{bmatrix} \Omega_1 & \Omega_2 \end{bmatrix} = \begin{bmatrix} \tilde{V}\tilde{\Sigma}^{-1}\tilde{U}_1^* & \tilde{V}\tilde{\Sigma}^{-1}\tilde{U}_2^* \end{bmatrix}$,
   where $\tilde{U}_1$ and $\tilde{U}_2$ are defined by (10).
3. Compute the truncated SVD of matrix $\tilde{H} = \tilde{\Sigma}^{-1}\tilde{U}_1^*$
   $\tilde{H} = \bar{U}\check{\Sigma}\check{V}^*$,
   with truncation value $r$. Substitite $\tilde{H}$ into (24) to obtain
   $\Omega_1 = \tilde{V}\bar{U}\check{\Sigma}\check{V}^* = \check{U}\check{\Sigma}\check{V}^*$,
   which is the truncated SVD of $\Omega_1$.
4. Compute the reduced approximations of $A$ and $B$ :
   $\tilde{A} = \check{V}^*\check{A}\check{V} = \check{V}^*Y\check{U}\check{\Sigma}$,
   $\tilde{B} = \check{V}^*\check{B} = \check{V}^*Y\Omega_2$,
5. Perform the eigenvalue decomposition of $\tilde{A}$
   $\tilde{A}W = W\Lambda$.
6. Compute the DMD modes of $A$
   $\Phi = Y\check{U}\check{\Sigma}W$.

---

It can be shown that Algorithm 3 identifies all of the nonzero eigenvalues of $A$. Suppose $A\phi = \lambda\phi$, for $\lambda \neq 0$, and let $\mathbf{w} = \check{V}^*\phi$. Then

$$\tilde{A}\mathbf{w} = \check{V}^*Y\check{U}\check{\Sigma}\mathbf{w} = \check{V}^*Y\check{U}\check{\Sigma}\check{V}^*\phi = \check{V}^*A\phi = \lambda\check{V}^*\phi = \lambda\mathbf{w}.$$

Furthermore, $\mathbf{w} \neq 0$, since if $\check{V}^*\phi = 0$, then $Y\check{U}\check{\Sigma}\check{V}^*\phi = A\phi = 0$ and $\lambda = 0$. Thus, $\mathbf{w}$ is an eigenvector of $\tilde{A}$ with eigenvalue $\lambda$, and is identified by *Algorithm 3* (and *Algorithm 2*).

*4.3. Computational Complexity*

It is worth pointing out that Algorithms 3 and Algorithms 2 are very similar. The main difference is that at Step 3, instead of the SVD of matrix $\Omega_1$ (in Algorithm 2), the SVD of matrix $\tilde{H}$, defined by (24), is used, plus one matrix multiplication. This leads to an improvement in the computational efficiency of Algorithm 3 over Algorithm 2. In Algorithm 2, we apply the SVD of a matrix with dimension $m \times n$, while in Algorithm 3, we use the SVD of a $p \times n$ matrix, where $p < m$. Next, we compare the computational complexity of the standard algorithm for the DMDc method (Algorithm 1) and the improved algorithm (Algorithm 3). The two algorithms contain the same number of steps and the computational complexities of the first two steps and the fifth step are comparable. The main difference is in calculating the SVD at Step 3 (in the two algorithms), see Table 1.

**Table 1.** Performed SVD at Step 3 in Algorithm 1 and Algorithm 3.

|  | Algorithm 1 | Algorithm 3 |
|---|---|---|
| SVD of matrix | $Y$ of size $m \times n$ | $\tilde{H}$ of size $p \times n$ |

The complexities of the corresponding matrices in steps 4 and 6 show the main difference; see Table 2.

**Table 2.** Reduced matrices and DMD modes.

|  | Algorithm 1 | Algorithm 3 |
|---|---|---|
| Reduced matrix | $\tilde{A} = \hat{U}^* Y \tilde{V} \tilde{\Sigma}^{-1} \tilde{U}_1^* \hat{U}$ | $\tilde{A} = \check{V}^* Y \check{U} \check{\Sigma}$ |
| DMD modes | $\Phi = Y \tilde{V} \tilde{\Sigma}^{-1} \tilde{U}_1^* \hat{U} W$ | $\Phi = Y \check{U} \check{\Sigma} W$ |

In Algorithm 1, six matrices need to be stored and five matrix multiplications need to be performed, while in Algorithm 3, it is necessary to store only four matrices and perform three matrix multiplications for the computation of the reduced matrix $\tilde{A}$. The same corresponding number of matrices and matrix multiplications are also required to calculate the DMD matrix $\Phi$ for the two algorithms, respectively.

**5. Numerical Illustrative Examples**

In this section, we will compare the results obtained using the standard DMDc algorithm and the new algorithm (Algorithm 3) introduced in Section 2. All considered examples are well known in the literature. All numerical experiments and simulations were performed on Windows 7 with MATLAB release R2013a on an Acer Aspire 571 G laptop with an Intel(R) Core(TM) i3-2328M CPU at 2.2 GHz and 4 GB of RAM. For the first two examples, numerical simulation results of the respective models were obtained using the standard *ode45* MATLAB solver for ordinary differential equations (ODEs).

**Example 1.** *A simple population dynamics model.*

We consider the Lotka–Volterra system, a two-dimensional, weakly nonlinear dynamical system describing the interaction between two competing populations. These dynamics may represent two species in biological systems, competition in stock markets [30] and can be modified to study the spread of infectious diseases [31]. The dynamics of the prey and predator populations, $x_1$ and $x_2$, respectively, are given by

$$\left|\begin{array}{ll} \dot{x}_1 = & \alpha x_1 - \beta x_1 x_2 \\ \dot{x}_2 = & -\gamma x_2 + \delta x_1 x_2 + u, \end{array}\right. \tag{28}$$

where the parameters $\alpha, \beta, \gamma$ and $\delta$ represent the growth and death rates, the effect of predation on the prey population and the growth of predators based on the size of the prey population. The control input $u$ affects only the second state. The unforced system exhibits limit cycle behavior, where the predator lags the prey, and a critical point is $x_{eq} = \left(\frac{\gamma}{\delta}, \frac{\alpha}{\beta}\right)^T$, where the population sizes of both species are in balance. The control objective is to stabilize this fixed point.

We have used the following parameter values: $\alpha = \frac{1}{2}, \beta = \frac{1}{40}, \gamma = \frac{1}{2}$ and $\delta = \frac{1}{200}$. The time-step $\triangle t = 0.1$ and sinusoidal forcing with $u(t) = (2\sin(t)\sin(t/10))^2$ are used. The initial condition is $x_0 = (60, 50)^T$. Collected data consist of $n = 501$ snapshots, i.e., the data matrices $X$ and $Y$ are of the dimensions $2 \times 500$, and $\Gamma$ is of dimensions $1 \times 500$. Figure 1 depicts the dynamics.

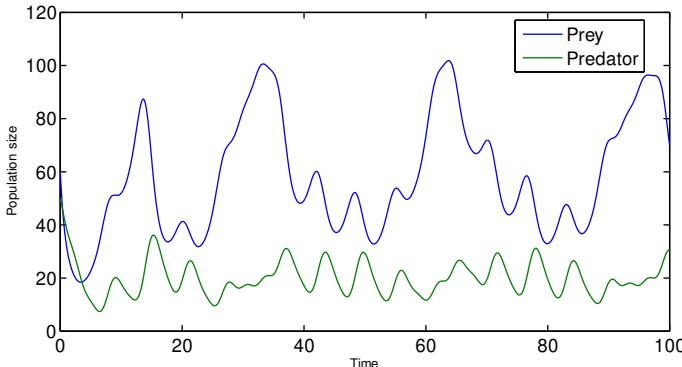

**Figure 1.** Lotka–Volterra dynamics (28) with $\alpha = \frac{1}{2}$, $\beta = \frac{1}{40}$, $\gamma = \frac{1}{2}$ and $\delta = \frac{1}{200}$, and initial condition $x_0 = (60, 50)^T$.

The low-dimensional representation (13) of the matrices $A$ and $B$, by Algorithm 1, are

$$\tilde{A} = \begin{pmatrix} 0.9989 & 0.0659 \\ 0.0049 & 0.9784 \end{pmatrix} \text{ and } \tilde{B} = \begin{pmatrix} -0.0190 \\ 0.0989 \end{pmatrix},$$

while the corresponding representations (20) by Algorithm 3 are

$$\tilde{A} = \begin{pmatrix} 0.9758 & -0.0040 \\ -0.0650 & 1.0015 \end{pmatrix} \text{ and } \tilde{B} = \begin{pmatrix} -0.0995 \\ -0.0152 \end{pmatrix}.$$

Both algorithms, Algorithms 1 and 3, produce the same DMD eigenvalues

$$\omega_1 = 0.9680 \text{ and } \omega_2 = 1.0094$$

and the same, up to the sign, corresponding DMD modes

$$\phi_1 = \begin{pmatrix} 0.7178 \\ 0.6494 \end{pmatrix} \text{ and } \phi_2 = \begin{pmatrix} 0.9996 \\ 0.1400 \end{pmatrix}.$$

In order to quantify the average magnitude of the residuals between estimated values obtained using the introduced DMDc approaches and actual data, we have used Root Mean Square Error (RMSE):

$$RMSE(i) = \sqrt{\frac{1}{n} \sum_{s=1}^{n} (x_i(s) - \hat{x}_i(s))^2}, \tag{29}$$

where $x$ are the actual values and $\hat{x}$ consists of the estimated values. The results are shown in Table 3.

**Table 3.** Root Mean Square Error (RMSE) computed by (29).

|  | Standard DMDc | Improved DMDc |
|---|---|---|
| RMSE(1) | $1.2276 \times 10^4$ | $0.3934 \times 10^3$ |
| RMSE(2) | $8.2040 \times 10^3$ | $0.4199 \times 10^3$ |

The Relative Error (RE) is another way to measure the quality of approximations:

$$RE(i, t) = \frac{\| x_i(t) - \hat{x}_i(t) \|}{\| x_i(t) \|} \tag{30}$$

for $t = 1, \ldots, 500$. Figure 2 shows a plot of the relative error variation.

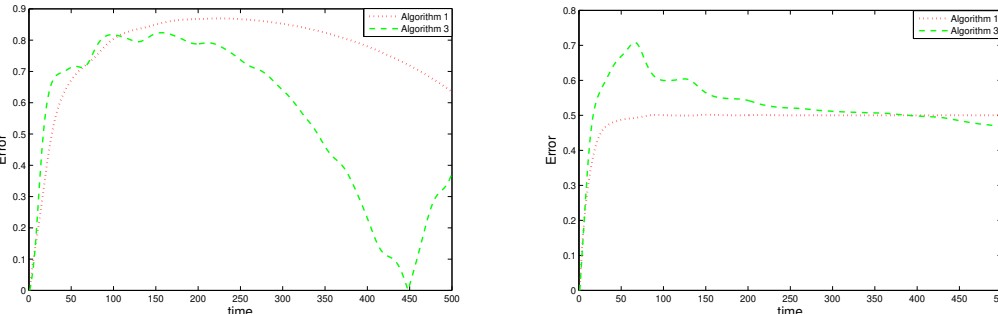

**Figure 2.** Relative errors: $RE(1, t)$ (**left panel**) and $RE(2, t)$ (**right panel**).

**Example 2.** *Chaotic Lorenz system.*

Let us consider the Lorenz dynamics given by

$$
\begin{vmatrix}
\dot{x}_1 = & \sigma(x_2 - x_1) + u \\
\dot{x}_2 = & x_1(\rho - x_3) - x_2 \\
\dot{x}_3 = & x_1 x_2 - \beta x_3
\end{vmatrix}
\tag{31}
$$

with system parameters $\sigma = 10$, $\beta = 8 = 3$, $\rho = 28$, and control input $u$ affecting only the first state. A typical trajectory oscillates alternately around the two weakly unstable fixed points $(\pm\sqrt{72}, \pm\sqrt{72}, 27)^T$.

The chaotic motion of the system implies a strong sensitivity to initial conditions, i.e., small uncertainties in the state will grow exponentially with time. The control objective is to stabilize one of these fixed points. The time-step $\triangle t = 0.001$ is used and control input is determined every 10 system timesteps and then held constant, and the actuation input is limited to $u \in [-50, 50]$. Collected data matrices $X$ and $Y$ are of the dimensions $2 \times 10,000$, and $\Gamma$ is of the dimensions $1 \times 10,000$. The initial condition is $x_0 = (-8, 8, 27)^T$. Figure 3 depicts the dynamics.

We performed Algorithm 1 and Algorithm 3 to obtain DMD eigenvalues and DMD modes. Two algorithms reproduce the same DMD eigenvalues $\omega_1 = 0.9996 + 0.0055i$, $\omega_2 = 0.9996 - 0.0055i$ and $\omega_3 = 0.9999$, see Figure 4.

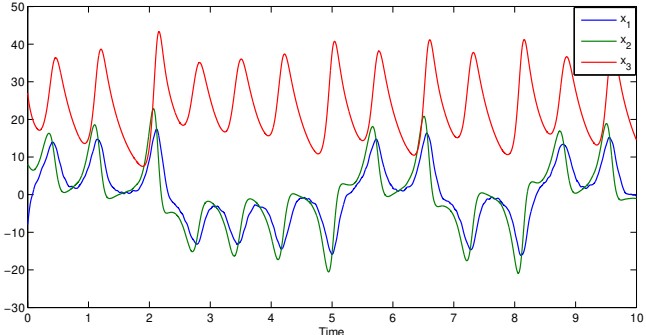

**Figure 3.** Lorenz dynamics (31) with $\sigma = 10$, $\beta = 8 = 3$, $\rho = 28$ and initial value $x_0 = (-8, 8, 27)^T$.

Figure 4 shows the DMD modes computed by two algorithms.

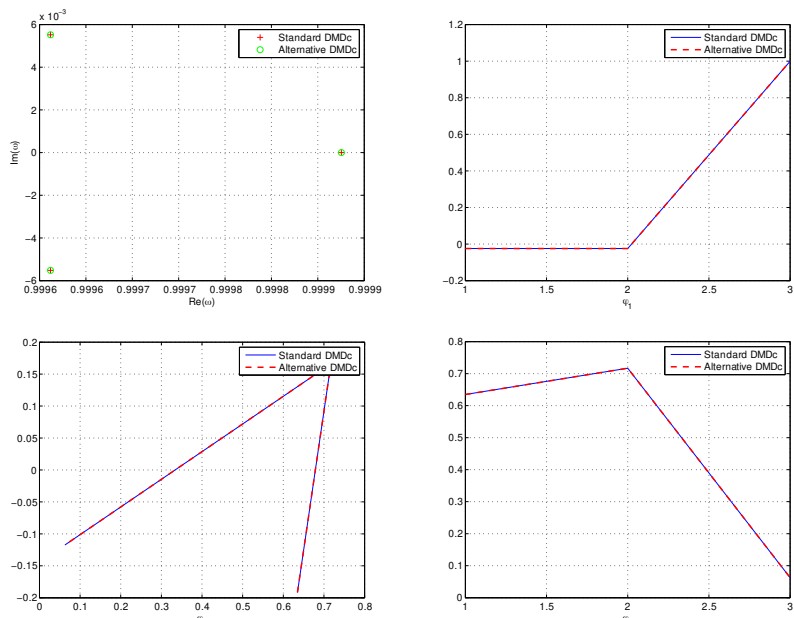

**Figure 4.** DMD eigenvalues (top-left panel) and DMD modes computed by Algorithms 1 and 3.

It can be seen from Figure 4 that the two algorithms produce the same DMD eigenvalues and DMD modes.

**Example 3.** *Large-scale, stable linear systems.*

Here we consider a large-scale dynamical system where the number of measurements is significantly larger than the dimensionality of the underlying system. The dynamics under consideration have an underlying low-dimensional attractor. To construct these large-scale systems, we use MATLAB's Discrete Random State Space Method. We have chosen a five-dimensional model with two input variables and 50 measurement variables. The output is a state space model: $A \in \mathbf{R}^{5 \times 5}$ and $B \in \mathbf{R}^{5 \times 2}$ (and $C \in \mathbf{R}^{50 \times 5}$, $D \in \mathbf{R}^{50 \times 2}$). We generate a matrix of random inputs, $\Gamma$, by using MATLAB's `randn` command. Using matrices $A$, $B$ and $\Gamma$, we generate output data for snapshot matrices $X$ and $Y$, which are of the dimensions $5 \times 50$, and $\Gamma$ is of dimensions $2 \times 50$.

The initial condition is $x_0 = (1, 1, 1, 1, 1)^T$. Figure 5 depicts the dynamics.

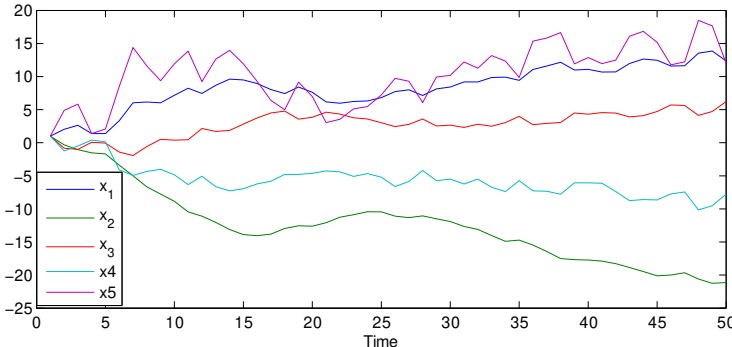

**Figure 5.** Dynamics of Example 3.

The DMD eigenvalues and modes calculated using Algorithms 1 and 2 are shown in Figure 6.

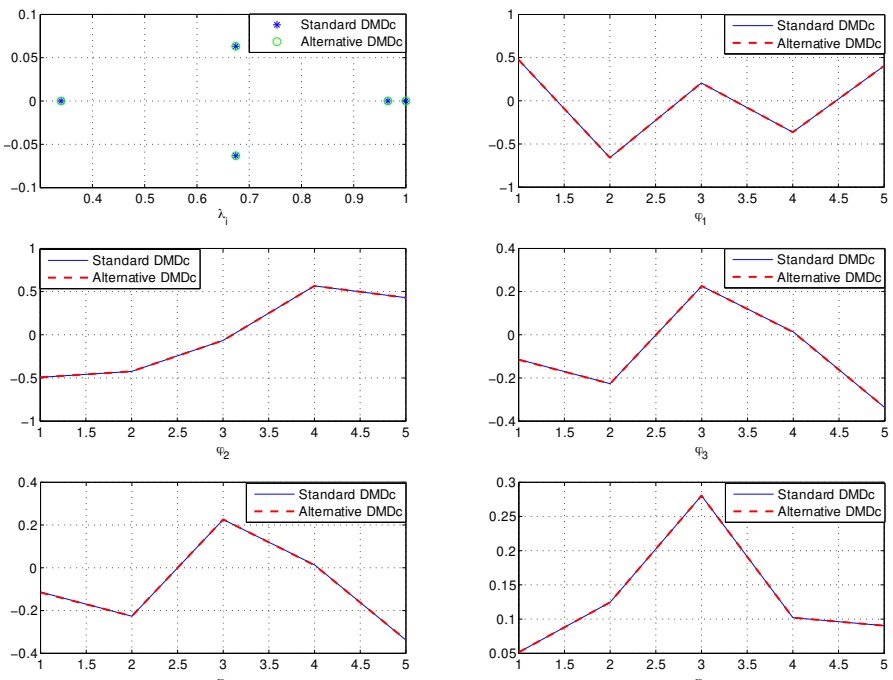

**Figure 6.** DMD eigenvalues (top-left panel) and DMD modes computed using Algorithms 1 and 2.

Note that there is no distinction between the DMD modes and eigenvalues generated by Algorithms 1 and 2, which is seen in all the examples considered.

## 6. Conclusions

In this study, we have presented a novel and efficient algorithm for implementing Dynamic Mode Decomposition with Control (DMDc), addressing the challenge of computational efficiency while maintaining predictive accuracy. We prove that each pairs $(\phi, \lambda)$ generated by Algorithm 3 is an eigenvector/eigenvalue pair of operator $A$ (Theorem 1). In addition, the new algorithm uses a lower-dimensional SVD (at Step 3) than the corresponding SVD in the standard algorithm. The low-dimensional representation of the matrices $A$ and $B$, and the matrix $\Phi$ of DMDc modes in Algorithm 3 has a simpler form than the corresponding matrices from Algorithm 1. Consequently, Algorithm 3 requires less memory and fewer matrix multiplications.

We have demonstrated the performance of the presented algorithms with numerical examples from different fields of application. From the obtained results, we can conclude that the introduced approach gives identical results to the standard DMDc method. The numerical results show that the introduced algorithm is an alternative to the standard DMDc algorithm and can be used in various fields of application.

The contributions of this work have several important implications for the field of DMDc and its applications. Firstly, the improved computational efficiency enables real-time analysis and control of complex systems, allowing for faster decision-making and response to changing environmental conditions. This aspect is particularly relevant in engineering applications, where rapid control strategies are crucial for system stability and performance. Furthermore, the accurate modeling of controlled systems opens up possibilities for advanced control and optimization techniques. By understanding the underlying dynamic modes and their responses to control inputs, researchers can design more effective and robust control strategies tailored to specific system behaviors and objectives. This has significant implications for engineering systems, ranging from aircraft and spacecraft control to advanced manufacturing processes and robotics.

While this study has made some progress in the realm of DMDc, there remain several exciting avenues for future research. One possible direction is integrating uncertainty

quantification into the DMDc framework. Uncertainties in measurement data and model parameters are common in practical applications, and accounting for these uncertainties can lead to more robust and reliable predictions. The development of uncertainty-aware DMDc methodologies could significantly enhance the algorithm's versatility and practicality in real-world scenarios. Additionally, investigating the application of our algorithm to control chaotic and turbulent systems could yield valuable insights into the predictability and controllability of such complex behaviors. Understanding how control inputs influence the long-term behavior of chaotic systems could have profound implications in various fields, from climate modeling to chaos control in mechanical systems. Finally, exploring the potential synergies between DMDc and other data-driven modeling and control techniques, such as Koopman operator theory and model predictive control, presents an exciting research opportunity. Integrating these methodologies could lead to novel hybrid approaches, combining the strengths of each technique to tackle complex and challenging control problems in an interdisciplinary manner.

**Funding:** This research received no external funding.

**Data Availability Statement:** No publicly archived data.

**Conflicts of Interest:** The authors declare no conflict of interest.

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
