# Peer review of "An Improved Approach for Implementing Dynamic Mode Decomposition with Control"

_computation, doi:10.3390/computation11100201_

Round 1

Reviewer 1 Report

The paper presents and interesting approach for the dynamic mode decomposition with control. The paperi is quite interesting but, in my opinion, some improvements are needed.

My remarks are listed as follows.

Abstract: it is suitable that authors do not present acronyms inside the abstract. Moreover, the abstract does not present the originality of the proposed research, as well as no details about the presented examples. I suggest the authors to revise the abstract completely.

Introduction: it is very weak and it is not suitable to address inexpert readers to the proposed problem. It does not contain the originality of the work and/or an exhaustive presentation of the main results. Moreover, the presentation of the mathematical details inside Introduction is not adequate. It appear only confusing. I suggest the authors to revise totally this section of the paper.

Section 2: I think it should be revised. The authors should define all the proposed article in a well-defined manner. Indeed, a part of Introduction (i.e. the subsection with mathematical details) should be inserted here.

Conclusion: they do not contain suitable future possible research activities. I suggest the authors to complete this part of the paper.

References: they are not adequate. In my opinion, some of them are useless for the presentation of the work. I suggest the authors to revise all references and to add only the ones that are really useful for the presentation of the manuscript.

In short, my final decision is “major revisions”.

A careful reading is necessary as some sentence are too long and, for this reason, they appear obscure.

Author Response

Based on yours' recommendations, we have revised the document to address the issues raised. The main changes concern the contents of the Abstract, Introduction, Conclusion and References sections. We hope these changes address the concerns adequately. 

Please see the attached file for the list of changes made.

Reviewer 2 Report

The paper is devoted to possible realizations of the dynamic mode decomposition for dynamical problems with controls. A minor modification of the algorithm is proposed in Algorithm 3, Step 3. This variant might require slightly more operations for SVD of \Omega_1 for the same threshold. In any case, this modification is not essential. The provided results for three test cases achieved by different modifications are practically identical. I do not see sufficient novelty for publication. Overall, the paper is well written. However, it does not contain enough novelty.

Author Response

In order to achieve a better presentation of our research, as well as to respond to some reviewers’ recommendations, we have made a revision of the original document. The main changes concern the contents of the Abstract, Introduction, Conclusion and References sections. We hope these changes address the concerns adequately. 

Please see the attached file for the list of changes made.

Reviewer 3 Report

Dynamic mode decomposition is an important method for the analysis of dynamical systems and with control DMDc is able to input-output systems. The main goal of this article is to introduce an improved alternative to the currently available algorithm for calculating the DMDc mode. A new economical approach is introduced from a computational point of view. Several examples are given to  demonstrate the applicability of the introduced technique although the involved dimensions of these examples are not very high. The paper seems interesting to researchers in nonlinear dynamics and control. I recommend its publication after minor revision.

The Language is good in general and the authors are encouraged to check typos or small wording problems.

Author Response

(The authors gave the same response as above.)

Reviewer 4 Report

1)     After formula (11) in the text "In DMDc, unlike DMD," please replace "unlike" with "in contrast with" (more formal construction).

Author Response

Dear Reviewer,

Thank you for your insightful review and constructive feedback. Based on your recommendations, we have revised the document to address the issues raised. Please see the attached file for point by point responses to your comments.

Round 2

Reviewer 1 Report

At the actual state, the paper deserves publication.

Author Response

Dear Reviewer,

We appreciate your time and effort reviewing our manuscript. Thank you for your decision. 

Reviewer 4 Report

Thank you for your detailed comments and the interesting manuscript. In my opinion, it has been improved and the current version is suitable for publication. Just in case, please check the text in the Figure 2 legend - shouldn't there be "Algorithm 3" instead of "Algorithm 2" (since algorithms 1 and 3 were compared before).